# Epstein–Barr Virus BRRF1 Induces Butyrophilin 2A1 in Nasopharyngeal Carcinoma NPC43 Cells via the IL-22/JAK3-STAT3 Pathway

**DOI:** 10.3390/ijms252413452

**Published:** 2024-12-15

**Authors:** Yue Liu, Ka Sin Lui, Zuodong Ye, Luo Chen, Allen Ka Loon Cheung

**Affiliations:** 1Medical School, Fuyang Normal University, Fuyang 236000, China; 202312002@fynu.edu.cn; 2Department of Biology, Faculty of Science, Hong Kong Baptist University, Kowloon Tong, Hong Kong SAR, China; kasinlui@hkbu.edu.hk (K.S.L.); zuodongye2022@163.com (Z.Y.); 3Department of Chemistry, Faculty of Science, Hong Kong Baptist University, Kowloon Tong, Hong Kong SAR, China; chenluo2cn@outlook.com

**Keywords:** butyrophilin, Epstein–Barr virus, BRRF1, nasopharyngeal carcinoma, gamma-delta T cells, IL-22

## Abstract

Epstein–Barr virus is highly associated with nasopharyngeal carcinoma (NPC) with genes expressed for tumor transformation or maintenance of viral latency, but there are certain genes that can modulate immune molecules. Butyrophilin 2A1 (BTN2A1) is an important activating protein for presenting phosphoantigens for recognition by Vγ9Vδ2 T cells to achieve antitumor activities. We have previously shown that Vγ9Vδ2 T cells achieve efficacy against NPC when BTN2A1 and BTN3A1 are upregulated by stimulating EBV gene expression, particularly LMP1. While BTN3A1 can be induced by the LMP1-mediated IFN-γ/JNK/NLRC5 pathway, the viral gene that can regulate BTN2A1 remains elusive. We showed that BTN2A1 expression is directly mediated by EBV BRRF1, which can trigger the BTN2A1 promoter and downstream JAK3-STAT3 pathway in NPC43 cells, as shown by RNA-seq data and verified via inhibitor experiments. Furthermore, BRRF1 downregulated IL-22 binding protein (IL-22RA2) to complement the EBNA1-targeting probe (P_4_)-induced IL-22 expression. Therefore, this study elucidated a new mechanism of stimulating BTN2A1 expression in NPC cells via the EBV gene BRRF1. The JAK3-STAT3 pathway could act in concordance with IL-22 to enhance the expression of BTN2A1, which likely leads to increased tumor cell killing by Vγ9Vδ2 T cells for enhanced potential as immunotherapy against the cancer.

## 1. Introduction

Nasopharyngeal carcinoma (NPC) is a type of malignant cancer that occurs in the nasopharynx. The etiological factor of NPC is not fully understood, but associations with Epstein–Barr Virus (EBV), alcohol, smoking, and diet have been made, such as cured fish [1]. There is a higher incidence of NPC in regions in southern China, including Hong Kong and Taiwan, as well as Inguiny in Greenland [2]. Usually, diagnosis of NPC occurs at the late stages when symptoms become obvious, and classical surgery, chemotherapy, and/or radiotherapy are not effective in eliminating the cancer [3]. The heterogeneity of NPC, the alterations of cellular pathways, and the diversity of the immune cells found in the tumor microenvironment adds to the challenge [4,5]. Therefore, the development of immunotherapies, such as the use of immune cells, has increasing potential for treating NPC patients when other options fail [6]. However, for the success of such new therapies, the understanding of the prevailing EBV within NPC and whether its viral genes have any relationship with promoting antitumor immune responses is one avenue to explore.

Some latent EBV-encoded proteins, such as EBV-encoded nuclear antigen 1 (EBNA1), are persistently expressed and can promote tumorigenesis. EBNA1 is found in all EBV latently infected cells and thus can be used as a target for manipulation [7]. We, and others, have previously used EBNA1-targeting probes (L_2_)P_4_ and ZRL_5_P_4_, which emit fluorescent signals when they bind with the EBNA1 protein, to stimulate reactivation of the latent viral genome [8,9,10]. Specifically, we found that the (L_2_)P_4_ probe can trigger EBV LMP1 expression, which promotes the expression of butyrophilins (BTNs) essential for activating the antitumor cytotoxic function of a type of innate immune T-cell named Vγ9Vδ2 T cells. In addition to latent membrane protein 1 (LMP1), other EBV genes could still be associated with BTN expression, such as BRRF1, as indicated by our previous analysis [11].

BRRF1 is not a well-characterized gene. It is dispensable for viral replication and encodes a transcription factor function [12,13]. The gene is encoded in the same region as the BZLF1 and BRLF1 EBV gene products [14,15]. However, its methylated promoter can be activated by BZLF1 during lytic replication [16]. BRRF1 enhances BRLF1-induced reactivation in gastric cancer cells and HEK293 cells. Knockdown of endogenous BRRF1 inhibits chemical-induced reactivation in NPC cell lines [17]. BRRF1 was also shown to induce cleaved protein expression [17]. It has been suggested that BRRF1 can interact with TRAF2 and mediate several major signaling pathways, including NF-κB and JNK activation [18]. In EBV-infected B cells, TRAF2 interacts with LMP1, which activates the NF-κB pathway [19]. In addition, studies have shown that BRRF1 and the p53 tumor-suppressing protein can work together to drive lytic gene expression within the epithelium [20]. BRRF1 can also act as a transcription factor by promoting the transcription function of c-Jun and activating the BZLF1 promoter through the cAMP response element (CRE) motif sequence [13]. From these studies, the function of BRRF1 has been attributed to mediating host cell pathways or other EBV genes that are related to EBV infection; however, there has been no indication that BRRF1 can affect the expression of BTN molecules that affect Vγ9Vδ2 T cells.

Vγ9Vδ2 T cells are a subtype of human T cells that have natural tumoricidal activities [21]. They are primarily activated in a T-cell receptor (TCR)-dependent manner by the recognition of phosphoantigen (pAg) presented by the complex formed between BTN2A1 and BTN3A1 molecules [22,23]. BTNs are a family of transmembrane glycoproteins that play important roles in the modulation of cellular and immune responses [24]. The intracellular B30.2 domain of BTN3A1 is essential for recognizing pAg as well as activating Vγ9Vδ2-TCR [25], where its configuration and transport on the cell membrane are mediated by Rho GTPase such as RhoB [26]. Recent studies have shown that BTN2A1 binding to BTN3A1 is the rate-limiting factor and vital for Vγ9Vδ2 T-cell cytotoxicity [27], where BTN2A1 binds to the Vγ9 chain of the TCR directly [22]. Thus, BTN2A1 and BTN3A1 molecules are particularly important for consideration in studying the effects of Vγ9Vδ2 T cells in immunotherapy against cancer. There have been several clinical trials that made use of these cells against different cancers, including NPC [28], but the limited effectiveness is underexplored in the realm of BTN expression in tumors. In this regard, we recently demonstrated that Vγ9Vδ2 T cells had enhanced efficacy against NPC when the EBNA1-targeting probe (P_4_) was used in conjunction with an in vivo mouse model [11]. Furthermore, we found that P_4_ resulted in the expression of LMP1, which led to the activation of the IFN-γ/p-JNK/NLRC5 axis and induced a certain level of BTN2A1/BTN3A1 expression in the tumors [11]. Given that BTN2A1/BTN3A1 is vital for the success of Vγ9Vδ2 T-cell immunotherapy against NPC, thus, we continued to explore if other EBV genes can directly influence their expression.

In this study, we uncovered a new function of EBV BRRF1, where it can stimulate the expression of BTN2A1 in NPC cells. BRRF1 does so by triggering the promoter activity of BTN2A1, leading to increased expression levels. In addition, we showed that while the EBV-reactivating agent P_4_ could stimulate IL-22 cytokine expression in NPC43 cells, BRRF1 can downregulate the level of the inhibitory IL-22 binding protein. Together, BRRF1 plus IL-22 can induce the highest level of BTN2A1 expression in NPC43 and HK1-EBV cells. As a result, IL-22 is potentiated to activate the downstream JAK3-STAT3 pathway to achieve a greater expression of BTN2A1 on the NPC tumor cells.

## 2. Results

### 2.1. BRRF1 Promotes the Expression of BTN2A1 and BTN3A1 in NPC43 Cells

Based on our previous report, we found that BRRF1 may be associated with the expression of BTN2A1 and BTN3A1 in NPC cells [11]. To investigate this possibility, we first overexpressed BRRF1 in EBV-infected cell lines (HK1-EBV and HONE1-EBV), clinically derived NPC cell lines (C666-1 and NPC43) [29], and gastric cancer cell lines (AGS, NCC-24 and SNU719), and detected the gene expression of *BTN2A1* and *BTN3A1*, via RT–qPCR (Figure 1A–D and Appendix A). In all of these cell lines, we found that *BTN2A1* was significantly induced by BRRF1 at 6 h and 24 h time points, with the highest induction observed in NPC43 followed by HK1-EBV (Figure 1A,B). *BTN3A1* had a significant increase only in NCC-24 at 6 h, and AGS and HK1-EBV at 24 h post transfection (Figure 1C,D). Western blot analysis confirmed that protein expression of BTN2A1, but not BTN3A1, was significantly upregulated in NPC43, HK1-EBV, and NCC-24 cells (Figure 1E–G).

Next, to test whether BRRF1 could directly trigger the transcription of these genes, a luciferase reporter assay was performed in HEK293T cells using plasmids encoding the promoters of *BTN2A1* and *BTN3A1*. Interestingly, BRRF1 triggered *BTN2A1* promoter activity but not *BTN3A1* promoter activity (Figure 1H). Based on these results, it appears that BRRF1 can stimulate the expression of BTN2A1 in cells, which has not been shown before.

### 2.2. Analysis of the Transcriptome Following BRRF1 Expression in NPC43 Cells

To investigate the mechanism of BRRF1 induction of BTN2A1, we overexpressed BRRF1 in NPC43 cells and performed RNA sequencing (RNA-seq). NPC43 was chosen because among the cell lines tested, both the BTN2A1 gene and protein expression were consistently upregulated; it is a clinically derived NPC cell line with features such as persistence of the EBV episome and viral gene expression, with retained 94% missense, stopgain, splicing, insertions, and deletions (INDELs) between the cell line and the original patient’s NPC [29]. Among the top 47 differentially expressed genes, we found 18 significantly upregulated genes and 29 downregulated genes compared to the control (Figure 2A). Some of these genes appear to be related to Ras GTPase binding and signaling, extracellular structure, and MAPK or PI3K-Akt signaling pathways based on KEGG and GO enrichment analyses (Figure 2B,C). These genes will serve as the basis to investigate the mechanism of BRRF1 function related to BTN2A1 induction.

We selected genes of interest based on the available information of their functions that may be related to transcriptional control or signaling pathways. Eleven upregulated genes (Appendix A) and nine downregulated genes (Appendix A) were chosen for RT–qPCR verification. Indeed, overexpression of BRRF1 in NPC43 upregulated the expression of the *JAK3* and *RhoB* genes while downregulating the *IL22RA2* gene (Figure 3A). The other genes showed no statistically significant differences (Appendix A). Via Western blotting, overexpression of BRRF1 in NPC43 cells after 24 h led to increased JAK3 and decreased IL-22RA2 protein expression, with minimal effect on RhoB expression (Figure 3B).

IL-22RA2 (or IL-22 binding protein, IL-22BP) is a naturally secreted monomeric protein whose primary function is to block the interaction of IL-22 with IL-22R by binding specifically to extracellular IL-22 and thereby inhibiting downstream cellular responses [30]. IL-22, IL-22R, and IL-22RA2 interactions play an important role in health and diseases and are involved in regulating cancer and homeostasis [31]. Therefore, the downregulation of *IL22RA2* mRNA and protein in NPC43 cells caused by BRRF1 likely promotes IL-22 binding with IL-22R (Figure 3A,B). In addition to the decreased expression of *IL22RA2*, we wondered whether BRRF1 can stimulate the expression of *IL22*. However, RT–qPCR analysis showed that BRRF1 did not directly induce gene expression of *IL22*, but stimulation of NPC43 cells using P_4_ could do so (Figure 3C). This suggests that EBV reactivation can induce the expression of *IL22* and may be mediated by EBV genes other than BRRF1. Moreover, *BTN2A1* gene expression was increased when we overexpressed BRRF1 in NPC43 and HK1-EBV cells, and IL-22 treatment appears to have a cooperative effect to further increase the expression of *BTN2A1* (Figure 3D,E). However, BRRF1 did not influence RhoB protein expression (Figure 3B), indicating that the Ras-GTPase pathway is not a major target of BRRF1.

### 2.3. BRRF1 Induces BTN2A1 Expression via the JAK3-STAT3 Pathway

Consistent with the RNA-seq and RT–qPCR data, BRRF1 increased JAK3 protein expression level (Figure 3B), suggesting that the JAK signaling cascade pathway could be activated by BRRF1. To test the other genes belonging to this family, BRRF1 also induced higher mRNA expression of *JAK1* and *TYK2* but not *JAK2* (Appendix A). Furthermore, an inhibitor against JAK3, ritlecitinib, significantly decreased the gene expression level of *BTN2A1* in NPC43 cells (Figure 3F). In addition, IL-22 synergistically upregulated *BTN2A1* promoter activity with BRRF1, and this effect was abrogated when JAK3 was inhibited (Figure 3G). At the protein level, BRRF1 transfection in NPC43 cells led to increased JAK3 and p-JAK3 expression but not TYK2/p-TYK2, JAK1/p-JAK1, and JAK2/p-JAK2, which confirmed that the JAK3 pathway plays an important role in BRRF1-induced BTN2A1 expression in NPC43 cells (Figure 4A and Appendix A). Inhibition of JAK3 corroborated with this finding. Ritlecitinib-treated NPC43 cells can decrease JAK3/p-JAK3 and successfully downregulate the effect of BRRF1 induction on BTN2A1 protein levels. No significant effects were shown for the other JAK family proteins. Interestingly, JAK3 inhibition appears unable to alter the expression of BTN3A1, indicating that BRRF1 induction of BTN3A1 may be dependent on other pathways.

Upon binding to IL-22R1, IL-22 signals via the JAK-STAT pathways where STAT1, STAT3, and STAT5 are primarily activated [32]. To further examine the effect of BRRF1 in cooperation with IL-22, we measured the protein expression of STAT3, STAT1, and STAT5 in NPC43 cells under the influence of BRRF1 overexpression with or without ritlecitinib. As shown in Figure 4B, BRRF1 induced STAT3 and p-STAT3 protein expression levels but not STAT1/p-STAT1, STAT5/p-STAT5, or JNK/p-JNK. Ritlecitinib abrogated this effect, which indicates that JAK3 is responsible for activating STAT3 for downstream induction of BTN2A1 (Figure 4B and Appendix A).

As well as the use of the JAK3 inhibitor, we confirmed the findings via siRNA experiments against JAK3 and STAT3. As shown in Figure 5A,B, si-JAK3 and si-STAT3 successfully reduced the gene expression of *JAK3* and *STAT3* in both NPC43 and HK1-EBV cell lines. For NPC43, treatment with si-JAK3, si-STAT3 or si-JAK3/si-STAT3 led to the decrease in *BTN2A1* expression following BRRF1 overexpression or in combination with IL-22 (Figure 5C). However, they did not affect IL-22-induced *BTN2A1* expression significantly (Figure 5C). A similar trend was seen with HK1-EBV, where downregulation of JAK3 and STAT3 by siRNA could also decrease the *BTN2A1* expression stimulated with BRRF1, IL-22 and BRRF1/IL-22 (Figure 5D). However, si-STAT3 did not have a significant effect against IL-22 treatment. In both cell lines, *BRRF1* expression was not significantly affected by the siRNA (Appendix A). We also examined the effect at the protein level via Western blotting (Figure 5E). In NPC43 cells, BRRF1, IL-22, and BRRF1/IL-22 can stimulate the increased protein expression of BTN2A1 when compared to β-actin control (Figure 5F). In comparison IL-22 alone did not have such effects in HK1-EBV. For both cell lines BRRF1/IL-22 induced the highest BTN2A1 protein expression. However, when si-JAK3 was used, the level of BTN2A1 proteins stimulated by the treatments were decreased in both cell lines, similar to si-STAT3 treatment (Figure 5E). Knockdown of both si-JAK3 and si-STAT3 led to a significant decrease in the level of BTN2A1 induced by BRRF1/IL-22 for both cell lines. These data corroborated with the JAK3 inhibitor findings above that BRRF1 and IL-22 cooperate through the JAK3/STAT3 pathway to trigger the expression of BTN2A1 in these NPC cell lines.

## 3. Discussion

During latency, gene expression by EBV remains subtle, and viral persistence is maintained by latent proteins such as EBNA1 and LMP1, which could also lead to tumor transformation of host cells [33]. Although EBV genes in oncogenesis are well characterized, there is also an opportunity to study their role (if any) in promoting antitumor immune cell functions. The difficulty for current immunotherapy against NPC is overcoming the immunosuppressive tumor microenvironment. The use of immune checkpoint inhibitors or ways to enhance antitumor T-cell functions resulted in limited success [34]. Therefore, other avenues should be explored to resolve the underlying reasons for the suboptimal effect of immunotherapy. Along this line, studying the existing EBV genes expressed within NPC tumor cells could augment cell-based immunotherapy. Here, we demonstrated that the EBV gene BRRF1, which can be triggered by an EBV-targeting probe (P_4_), upregulates the expression of BTN2A1 with the potential to support Vγ9Vδ2 T-cell adoptive immunotherapy against NPC.

BRRF1 is classified as a lytic gene that encodes a 34 kDa protein and can be activated by the transactivator BZLF1 during EBV replication [15]. The protein has been found in both the nucleus and cytoplasm [15,17]. BRRF1 has also been shown to activate the BZLF1 promoter and its own expression [13,17]. Furthermore, it was shown that BRRF1 can activate promoters that contain p53, AP-1, CREB, and NF-κB response elements [12]. Therefore, BRRF1 has the possibility to trigger BTN2A1 expression, which is shown by our data. BRRF1 appears to directly stimulate *BTN2A1* promoter activity and lead to the increased expression of BTN2A1 protein. However, for BTN3A1, BRRF1 appears to decrease the promoter activities, which is contradictory to the unchanged gene or protein expression detected following overexpression experiments (Figure 1). It is possible that BRRF1 may have triggered the expression of other EBV transactivators, such as BZLF1, or subsequently LMP1, which then activates the transcription of BTN3A1 [11]. Whether this is the case will need to be confirmed. Also, the synergistic activation of BTN2A1 and BTN3A1 that can be achieved by the expression of multiple EBV proteins should be explored in future investigation. Interestingly, we found that BRRF1 can upregulate molecules that could increase cytotoxicity by Vγ9Vδ2 T cells, such as NKG2D ligands (ULBP3 and ULBP4), as well as adhesion molecules ICAM-1 and RAGE (Appendix A). Another function we demonstrated for BRRF1 is the downregulation of the inhibitory IL-22 binding protein (IL-22RA2), which was alluded to by our RNA-seq data. We also demonstrated that P_4_-induced IL-22 would benefit from this, where higher activation of the JAK3-STAT3 pathway can be accomplished to stimulate the expression of BTN2A1. These functions have not been previously attributed to BRRF1.

BTN2A1 is an important molecule in activating the function of Vγ9Vδ2 T cells against tumor cells [27]. BTN2A1 and BTN3A1 form a complex associated with phosphoantigens that is recognized by Vγ9Vδ2-TCR [35], which drives downstream signaling in cells [23]. In the NPC tumor microenvironment, the expression of these molecules is lacking [11], and this would render γδ T-cell adoptive therapy ineffective when they infiltrate into the tumor site. The data from this study show that BRRF1 that can induce BTN2A1 will be an important aspect to consider in attempts to enhance the tumoricidal activity of Vγ9Vδ2 T cells [11]. It has been shown that RhoB mediates BTN3A1 translocation on the cell membrane for Vγ9Vδ2-TCR recognition [26]. However, our data show that BRRF1 overexpression did not significantly induce RhoB gene or protein expression, which could mean that the main function is associated for BTN2A1. As far as we know, the induction pathway of BTN2A1 remains elusive, and our data suggest that the JAK3 and STAT3 pathway could be one possible candidate. While others have showed indirect association between STAT3 and BTN3A1 [36,37], BRRF1 could likely mediate BTN3A1 via the activation of the JAK3/STAT3 pathway, which explains why the BTN3A1 promoter showed inverse activity upon BRRF1 overexpression. Together, BRRF1 has the potential to be used as a tool to modify NPC tumors prior to the use of Vγ9Vδ2 T-cell immunotherapy.

The finding that IL-22 could stimulate BTN2A1 is also surprising. IL-22 is part of the IL-10 family of cytokines and can target non-hematopoietic cells, including epithelial cells and fibroblasts, and can affect tissues in the gut, lung, liver, pancreas, breast, skin, kidney, and thymus [38]. It can be produced by cells of the lymphoid lineage, including αβ CD4^+^ and CD8^+^ T cells, and γδ T cells; Th17 cells can produce both IL-22 and IL-17 and cooperatively enhance the expression of antimicrobial peptides [39,40,41,42,43]. However, the role of IL-22 in promoting or suppressing tumors remains controversial. In colon cancer, IL-22 could activate STAT3 and cause progression [44], but there is no association of IL-22 with the causative effect on tumor transformation [45]. On another note, IL-22RA2 has a regulatory role, as it is constitutively expressed in various tissues, but in cancer, it appears to be decreased, which can allow IL-22-induced tumor growth [46]. Notably, IL-22RA2 is constitutively expressed in various tissues, and it counteracts the inflammatory properties of IL-22 [47]. However, in cancer, IL-22RA2 was found to be decreased, which would then allow IL-22 tumorigenic effects [48]. In our study, we showed that IL-22 promotes the expression of BTN2A1, which has the potential to activate the tumoricidal activities by Vγ9Vδ2 T cells. Whether the use of P_4_ treatment of NPC cells can achieve physiological concentration of IL-22 for BTN2A1 induction remains to be investigated, as the amount produced is most likely less than tumor-resident lymphoid cells. However, when IL-22-producing Th17, NKT, and Th22 cells infiltrate into the tumor, the increased amount of IL-22 can possibly offset the effect of BTN2A1 and Vγ9Vδ2 T cells, which eventually leads to tumor progression [39,49,50]. If this is the case, then the effect of BRRF1 with the downregulated IL-22RA2 will amplify tumor progression induced by IL-22. Therefore, the effect of IL-22 should be considered in the future development of Vγ9Vδ2 T-cell therapy in NPC patients.

The heterogeneity of the nature of NPC suggests that there could be multiple mechanisms that can regulate the expression of certain immune molecules and tumor antigens [4,5,51]. Several studies have shown differences in histone deacetylase inhibitors or protein kinase C activators in treating different EBV-positive NPC cell lines [29,52,53,54]. Our data also demonstrated a discrepancy where BRRF1 can induce BTN2A1 protein expression in NPC43 and HK1-EBV but not in the other EBV-positive cells. As we showed that JAK3 and STAT3 likely mediates BTN2A1 expression, it is also possible that there are mutations in these genes. *JAK3* and *STAT3* genes had been reported to be different among the NPC cell lines [55], and in other types of cancer including natural killer T cell lymphoma [56], and non-small cell lung cancer [57]. Thus, it will be worthwhile to analyze the genes encoding the JAK3/STAT3 pathway. This could lead to the understanding on how BTN2A1 expression could be less or absent in naturally occurring tumors that could hinder the effectiveness of Vγ9Vδ2 T cells. Furthermore, it will be interesting to extend this and future studies into EBV-positive non-NPC cell lines, such as gastric tumor cells SNU719, NCC-24, and AGS, or Raji B cells to examine if using P_4_, BRRF1, and/or IL-22 can have broad application to successfully induce BTN2A1 to complement combined anticancer therapy using Vγ9Vδ2 T cells.

There are several limitations in our study. First, we have not comprehensively studied the mechanism of how BRRF1 and JAK3/STAT3 induces BTN2A1. It is suggested that BRRF1 and STAT3 can trigger the BTN2A1 promoter, but chromatin immunoprecipitation should be performed to show definitive and direct evidence. Second, our data suggest that the expression of the RhoB protein was not significantly altered by BRRF1, but colocalization, binding or functional studies (e.g., membrane mobility) should be performed to determine if BRRF1 could also enhance RhoB-directed BTN2A1/BTN3A1 dimer formation [26]. Third, combination of BRRF1 and LMP1 transduction could be attempted in various cancers to determine if they will achieve high expression of BTN2A1/BTN3A1 proteins, especially among those NPC cell lines unresponsive to BRRF1 alone. Fourth, despite focusing on BTN2A1/BTN3A1, we have not examined if the immune checkpoint molecules such as PD-L1/L2, LAG-3, TIM-3, TIGIT, and Siglec-10/CD24, could also be induced by the EBV genes [58,59], which would counteract the effect of BTN2A1/BTN3A1. Addressing these limitations will strengthen the understanding of the use of EBV genes to mediate Vγ9Vδ2 T cells activation and cytotoxicity in immunotherapy.

In summary, we uncovered a new mechanism to induce BTN2A1 in NPC via BRRF1 and/or IL-22, with the downregulation of IL-22BP and activation of the JAK3-STAT3 pathway. BRRF1, as well as LMP1 [11], may have uses in transducing other tumors for promoting BTN2A1 and BTN3A1 expression prior to the adoptive transfer of Vγ9Vδ2 T cells; this may be one path of enhancing immunotherapy. Thus, the new findings here can provide insights to exploit tumor-associated viral genes in various cancers for immunotherapy.

## 4. Materials and Methods

### 4.1. Cell Culture and Plasmid

NPC cells (HK1, HK1-EBV, HONE1, and HONE1-EBV) were cultured using DMEM (GIBCO, Waltham, MA, USA) supplemented with 10% fetal bovine serum (FBS; GIBCO). NPC cells derived from clinical specimens (C666-1 and NPC43) [29], and gastric cancer cell lines NCC-24 and SNU719 were cultured in RPMI-1640 medium (GIBCO) supplemented with 10% FBS. The culture medium for NPC43 was supplemented with the ROCK inhibitor Y27632 (4 µM, Cat. No. S1049, Selleckchem, Houston, TX, USA). Another gastric cancer cell line AGS was cultured in F12K medium (GIBCO) supplemented with 10% FBS. All cell culture media were supplemented with 1% penicillin/streptomycin (GIBCO). The cells were maintained at 37 °C in a 5% CO_2_ incubator.

The expression plasmid encoding BRRF1 contained an c-Myc tag synthesized by the IGE company. The vector pcDNA3.1 was used as a control in the experiments. An antibody against the Myc tag (Cat. No. 60003-2-lg, Proteintech, Chicago, IL, USA), was used to detect BRRF1 expression in the cells.

### 4.2. Isolation of Total RNA and Reverse Transcription Quantitative Real-Time PCR (RT–qPCR)

RNAiso PLUS (Cat. No. 9109, Takara, Tokyo, Japan) was used to isolate total RNA from the cells. cDNA synthesis from total RNA was performed using the Prime Script RT Reagent Kit (Cat. No. RR047A, Takara) according to the manufacturer’s instructions. cDNA was then used for qPCR analysis using TB Green^®^ Premix Ex Taq™ II (Ti RNase H Plus) (Cat. No. RR820A, Takara) according to the manufacturer’s instructions. The *GAPDH* gene was used as a housekeeping control for normalization in data analysis. Appendix A shows the gene primer sets used in the qPCRs.

### 4.3. RNA Sequencing

Total RNA samples were sent to Novogene (Beijing, China) to perform RNA sequencing based on the NovaSeq PE 150 platform. Three independent experimental samples from control and BRRF1-overexpressing cells were used for RNA sequencing analysis. Twelve G of raw data per sample were generated for standard bioinformatic analysis. The RNA sequencing raw data is available at https://www.ncbi.nlm.nih.gov/sra/PRJNA1188645 (accessed on 20 November 2024).

### 4.4. Cell Transfection

Cells were seeded one day before transfection. Cell culture media was changed to Opti-MEM media (Cat. No. 31985062, GIBCO) 1 h before transfection. The transfectant plasmid DNA and PEI reagent was prepared in Opti-MEM. Three microliters of linear PEI (1 mg/mL, Cat. No. 23966-1, Polysciences, Warrington, PA, USA) transfection reagent was used per 1 μg of plasmid DNA. The transfectant mixture was gently vortexed and left at room temperature for 20 min to allow DNA/PEI complexes to form. The control empty vector with PEI was used as a vehicle control. The transfectant mixture was added to the cells uniformly, and the media was changed to cell culture media after 5–6 h. The cells were incubated at 37 °C and 5% CO_2_ before being used for experiments.

### 4.5. Inhibitor Experiments

JAK3 signaling was inhibited by ritlecitinib (PF-06651600, Cat. No.: HY-100754, MedChemExpress, Monmouth Junction, NJ, USA) in the cell experiments. DMSO was used as a control. Cells for experiments were seeded overnight to achieve 80% confluence before treatment with ritlecitinib for one hour, followed by BRRF1 transfection and incubation for another 24 h before assessment.

### 4.6. Western Blotting Analysis

Protein extracts from cells were assessed for protein concentrations using the Pierce BCA protein assay kit (Thermo Fisher Scientific, Waltham, MA, USA). Equal amounts of protein lysates were added to SDS–PAGE gels, and electrophoresis was performed. The proteins were then transferred to PVDF (Millipore, Burlington, MA, USA) membranes using a wet transfer system. The membranes were incubated with 5% blocking-grade blocker non-fat milk (Cat no. 1706404, Bio-Rad, Hercules, CA, USA) plus 0.5% bovine serum albumin (BSA; Cat. No. A2153, Sigma-Aldrich, Burlington, MA, USA) in Tris-buffered saline with 0.1% Tween-20 (TBS-T) for one hour at room temperature. Appropriate primary and secondary antibodies were then used for the membranes (Appendix A), which may be cut into strips according to the protein of interest. Signals were visualized using Pierce ECL Western Blotting Substrate (Cat. No. 32106, Thermo Fisher Scientific), images were taken with ChemiDoc (Bio-Rad), and band intensities were analyzed using ImageJ version 2.14.0/1.54f.

### 4.7. Luciferase Reporter Assay

The BTN2A1, BTN3A1, and NLRC5 promoter regions were inserted into the pGL3 base plasmid to create luciferase reporter plasmids. As a negative control, the pGL3 plasmid vector was used. Sequencing was used to verify all constructs. The cytokine IL22 (10 ng/mL, Cat. No. 13059-HNAE, Sino Biological, Beijing, China) was used in certain experiments after transfection in some experiments.

HEK293T or NPC43 cells were preinoculated in 24-well plates and cotransfected with the luciferase reporter plasmid and the corresponding plasmids using PEI as described above. After 30–40 h of transfection, cell lysates were analyzed using the Dual-Luciferase^®^ Reporter Assay System (Promega, Madison, WI, USA) following the manufacturer’s instructions. The pRLTK (Promega) luciferase reporter was used as a control. A GloMAX^TM^ 96 microplate luminometer (Promega) was used to measure the bioluminescence.

### 4.8. siRNA Experiments

NPC43 and HK1-EBV cells were seeded overnight before siRNA transfection. A total of 50 nM siRNA of antiJAK3, STAT3, JAK3/STAT3, and siRNA negative control (NC) (Guangzhou IGE Biotechnology, Guangzhou, China) and Lipofectamine 3000 transfection reagent (Invitrogen, Waltham, MA, USA) were prepared in Opti-MEM, followed by incubation at room temperature for 20 min. After 48 h of siRNA transfection, cells were transfected with BRRF1 plasmid or control plasmid with Lipofectamine 3000 reagent. IL-22 (10 ng/mL, Cat no. 782-IL-010, R & D Systems, Minnneapolis, MN, USA) was also added to the cells for the experiment. Cells were harvested at 3 h for qPCR analysis, and 24 h for Western blot analysis.

### 4.9. Statistical Analysis

The data shown are the mean ± SEM of at least three independent experiments. Paired Student’s *t* test analysis of variance was used for statistical analysis unless otherwise stated. *p* < 0.05 indicates statistical significance.

## Figures and Tables

**Figure 1 ijms-25-13452-f001:**
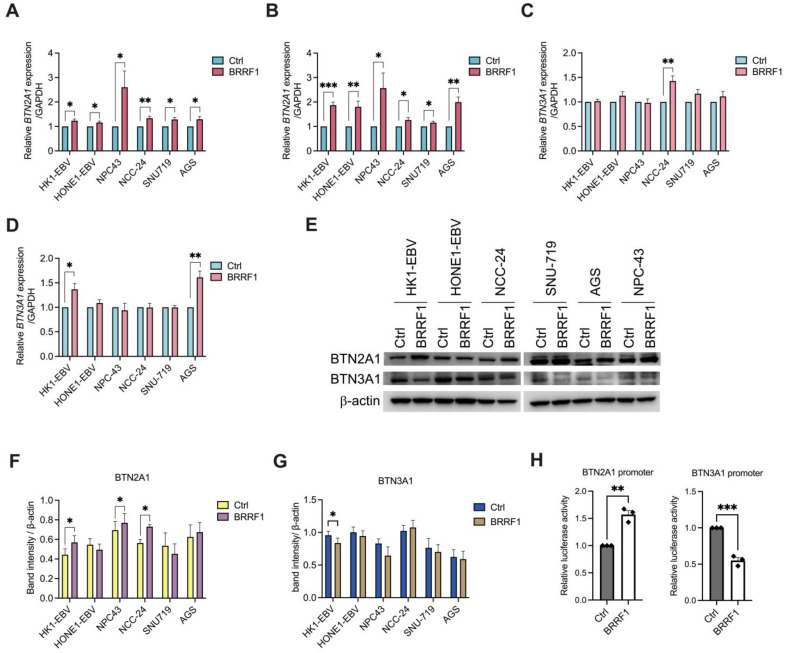
BRRF1 overexpression induced the expression of BTN2A1 and BTN3A1 in certain tumor cell lines. Cells were transfected with BRRF1 and assessed for *BTN2A1* and *BTN3A1* gene expression for 6 h (**A**,**B**) and 24 h (**C**,**D**), respectively, via RT–qPCR. (**E**) Western blot analysis of BTN2A1, BTN3A1, and β-actin protein levels of BRRF1 overexpression in different cell lines for 24 h compared to vehicle control. Representative immunoblots are shown. (**F**,**G**) Band intensities for BTN2A1 and BTN3A1 normalized to the β-actin for the different cell lines as column graphs are shown. (**H**) Luciferase activities from promoters for *BTN2A1* and *BTN3A1* in 293T cells cotransfected with plasmids encoding BRRF1 or vector control for 1 day. Data shown as the mean ± SEM from 3 independent experiments. Student’s *t*-test was performed for statistical analysis. * *p* < 0.05, ** *p* < 0.01, *** *p* < 0.001.

**Figure 2 ijms-25-13452-f002:**
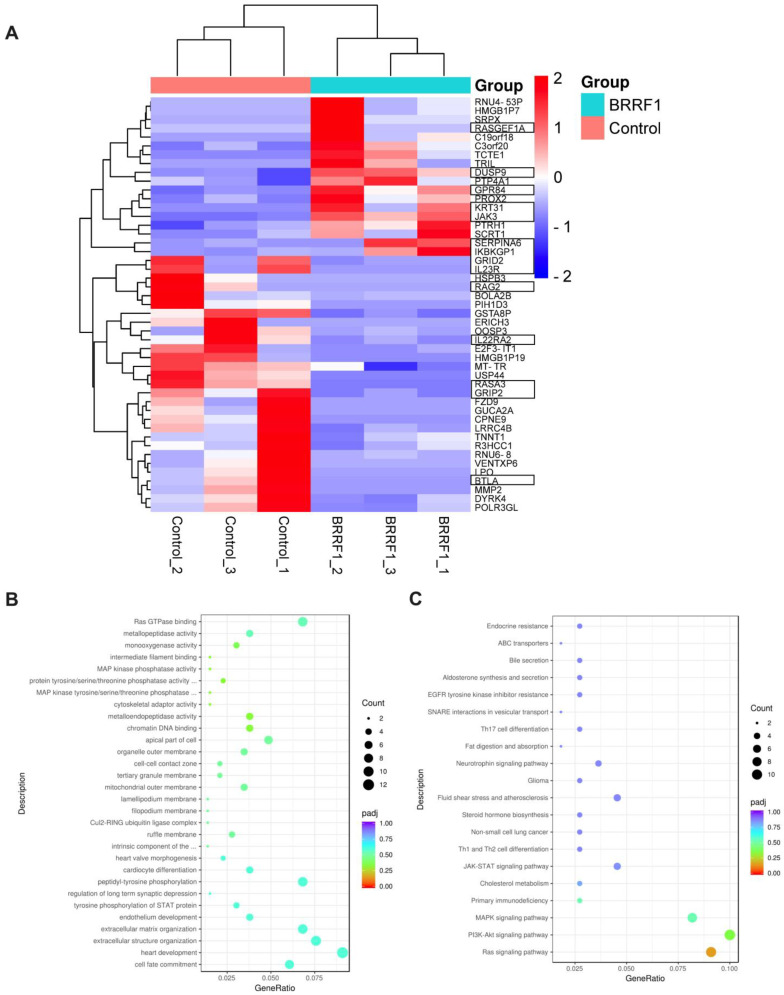
RNA sequencing analysis of genes altered by BRRF1 overexpression in NPC43 cells. (**A**) Heatmap of the RNA sequencing data in the control and BRRF1 overexpression groups. Three samples from each group were used for clustering. The degree of gene expression of BRRF1 compared to the control ranged from blue (downregulated) to red (upregulated) on a log_2_ scale. (**B**) KEGG and (**C**) GO analysis of the differentially expressed genes in BRRF1 versus control.

**Figure 3 ijms-25-13452-f003:**
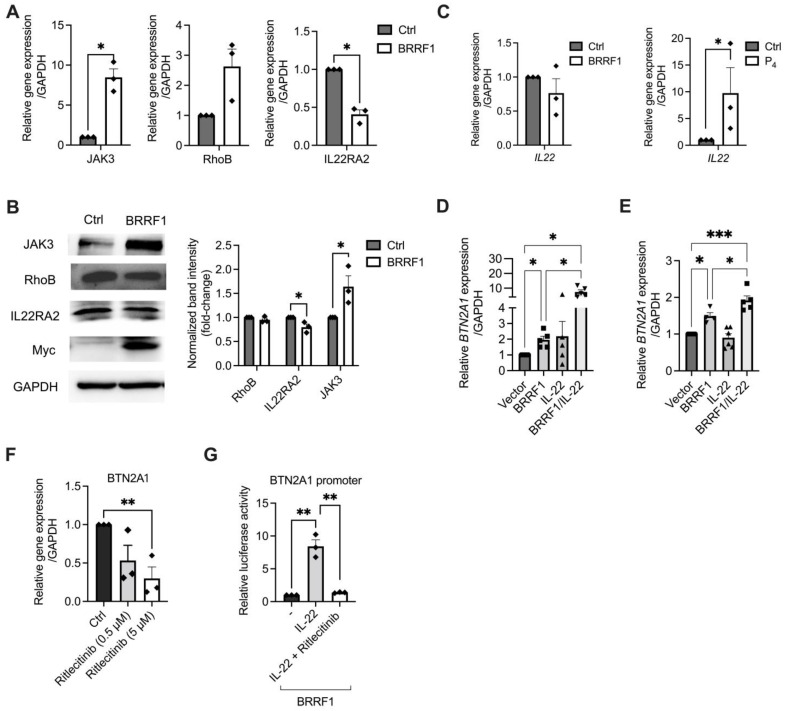
BRRF1 induces BTN2A1 expression through the IL22 and JAK3 pathways. (**A**) NPC43 cells were transfected with BRRF1 for 3 h and assessed for *JAK3*, *RhoB*, and *IL22RA2* gene expression via RT–qPCR. (**B**) Representative immunoblots from Western blot analysis of BRRF1 (Myc), RhoB, IL22RA2, and JAK3 expression following transfection of NPC43 cells by BRRF1 for 24 h. Band intensities are calculated and normalized to GAPDH. (**C**) NPC43 cells were transfected with BRRF1 or treated with 10 mM P_4_ and assessed for *IL22* gene expression via RT–qPCR. *BTN2A1* expression was assessed following BRRF1 overexpression with or without IL-22 treatment on (**D**) NPC43, and (**E**) HK1-EBV cells. (**F**) RT–qPCR of *BTN2A1* gene expression in NPC43 cells treated with the JAK3 inhibitor ritlecitinib (0.5 and 5 μM) or DMSO control. (**G**) Luciferase reporter assays using the *BTN2A1* promoter in NPC43 cells following BRRF1 overexpression with or without IL-22 (10 ng/mL) treatment. Column graphs represent data as the mean ± SEM from ≥3 independent experiments. Student’s *t* test was performed for statistical analysis. * *p* < 0.05, ** *p* < 0.01, *** *p* < 0.001.

**Figure 4 ijms-25-13452-f004:**
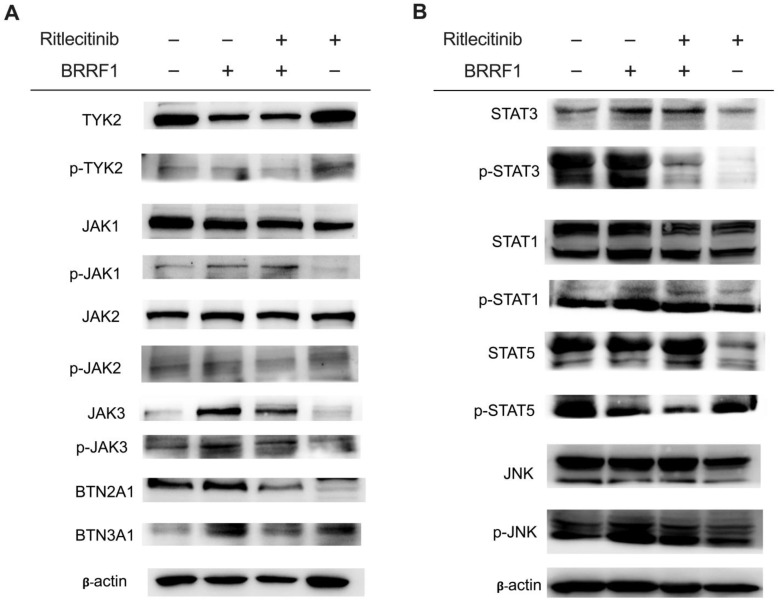
BRRF1 induces the BTN2A1 protein expression level through the JAK3-STAT3 pathway. (**A**) Western blot analysis of TYK2, p-TYK2, JAK1, p-JAK1, JAK2, p-JAK2, JAK3, p-JAK3, BTN2A1, and BTN3A1, as well as (**B**) STAT3, p-STAT3, STAT1, p-STAT1, STAT5, p-STAT5, JNK, and p-JNK protein expression of NPC43 cells by BRRF1 overexpression with or without ritlecitinib (5 μM) treatment. Representative immunoblots are shown.

**Figure 5 ijms-25-13452-f005:**
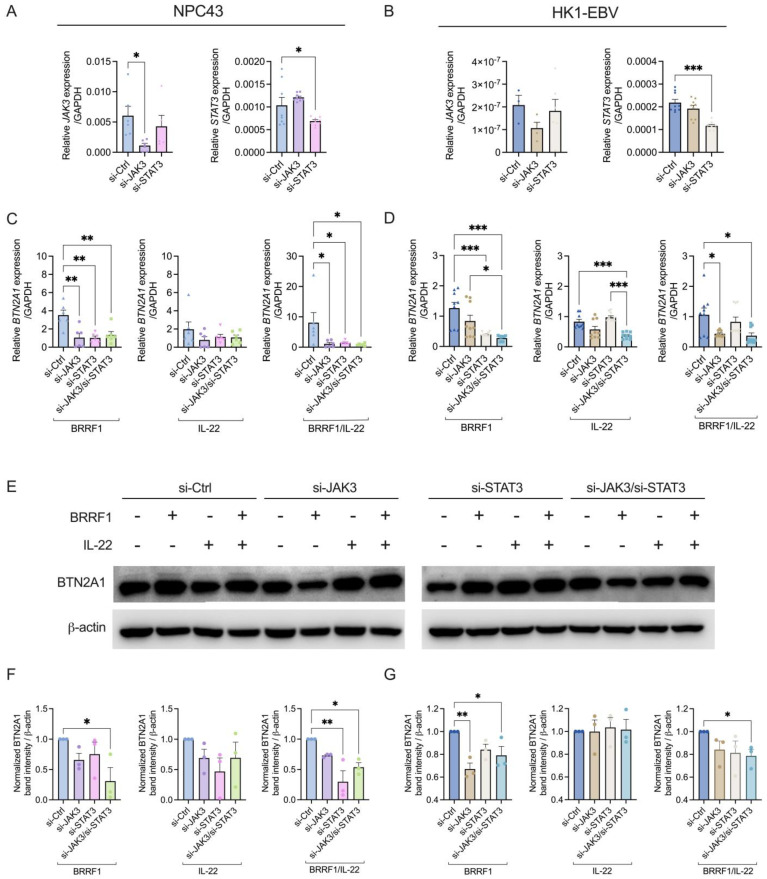
BTN2A1 expression is reduced when JAK3/STAT3 is suppressed. Expression of *JAK3* and *STAT3* was assessed following transfection with siRNA against JAK3 or STAT3, or scrambled control (si-Ctrl) in NPC43 (**A**) and HK1-EBV (**B**) cells. RT-qPCR analysis of *BTN2A1* expression under BRRF1 overexpression and/or IL-22 treatment in (**C**) NPC43 or (**D**) HK1-EBV cells following siRNA transfection for 48 h. Under these treatments, protein expression of BTN2A1 in NPC43 and HK1-EBV was assessed via Western blot analysis. (**E**) The representative immunoblots for HK1-EBV are shown, or as band intensities for (**F**) NPC43 and (**G**) HK1-EBV under different treatments. Data represents mean ± SEM from at least 3 independent experiments. ANOVA was performed for statistical analysis. * *p* < 0.05, ** *p* < 0.01, *** *p* < 0.001.

## Data Availability

The datasets analyzed during the current study are available from the corresponding author upon reasonable request.

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
