# Peer review of "Epstein–Barr Virus BRRF1 Induces Butyrophilin 2A1 in Nasopharyngeal Carcinoma NPC43 Cells via the IL-22/JAK3-STAT3 Pathway"

_ijms, 2024, doi:10.3390/ijms252413452_

Round 1
Reviewer 1 Report (Previous Reviewer 1)
Comments and Suggestions for Authors
Despite the efforts made to improve the manuscript, the main question raised in the first review remains unanswered. The results, although sometimes confusing and not always convincing, remain limited to a specific cell type and cannot be generalised. In this sense, the title of the manuscript is misleading. Some controls are missing (BRRF1 expression should be checked by Western blot in each transfection, etc.). Many results, sometimes contradictory, but no answer to the basic question: Why the specificity of NPC-43 cells?
Comments on the Quality of English LanguageCan be improved
Author Response
Comment 1:
Despite the efforts made to improve the manuscript, the main question raised in the first review remains unanswered. The results, although sometimes confusing and not always convincing, remain limited to a specific cell type and cannot be generalised. In this sense, the title of the manuscript is misleading. Some controls are missing (BRRF1 expression should be checked by Western blot in each transfection, etc.). Many results, sometimes contradictory, but no answer to the basic question: Why the specificity of NPC-43 cells?
Response 1:
Thank you for the comments. Regarding the results, we have revised the presentation of the data in figures that are ambiguous and makes it confusing. Figure S1, which was showing data from the first version of the manuscript has been removed. The data described in this Figure S1 was redundant when it also examined BTN2A1 and BTN3A1 expression, which is already shown in the revised Figure 1. The Figure S1 also included the NLRC5 data that was more relevant to BTN3A1 as shown from our previous study in Theranostics (Liu et al., 2023). We think this data would be confusing to the reader and derailed from the focus on understanding BTN2A1 expression induced by BRRF1.
The other problem was the previous Figure 5 F-G for the siRNA experiments. The presentation of the graphs as a whole with all the treatments made it difficult to understand the data. We have revised them to be consistent to the gene expression data shown in Figure 5 C-D to better present the data from the siRNA experiments. Following this point, we understand that there is discrepancy between the gene expression data and the protein expression data, this is most likely due to the level of gene expression not altered enough to be reflected at the protein level even though statistical significance was made. It is possible that some post-transcriptional regulation may have affected the expression of the proteins following BRRF1 induction.
In terms of the limit of using NPC43, we also used HK1-EBV cell lines to show that BRRF1 or BRRF1/IL-22 can induce the expression of BTN2A1 (Figure 3E). This cell line was also used in the siRNA experiments described in Figure 5. Both NPC43 and HK1-EBV was used because overexpression of BRRF1 lead to the upregulation of BTN2A1 gene and protein expression (Figure 1 A-B and F). These data were included in the previous submitted version of the manuscript.
However, for the RNA-seq data described in Figure 2, and most of the data in Figure 3, we still used NPC43. NPC43 is a cell line derived from clinical patient’s NPC (Lin et al., Nat Commun 2018). Compared to other patient-derived cell lines, it was found to be persistent with EBV episome and viral gene expression even after repeated culture. It also retained 94% missense, stopgain, splicing, insertions and deletions (INDELs) between the cell line and the original patient’s NPC (Lin et al., Nat Commun 2018). These features are important for trying to understand NPC, which is regarded as a type of cancer with high heterogeneity. The HONE1-EBV and HK1-EBV cell lines are lab-established, while C666-1 that we previously examined in the initial manuscript, is a clinical isolate that had been extensively used in the lab since 1999. Therefore, NPC43 is the most recently established clinical NPC cell line that is probably the best choice for understanding the interaction between EBV genes and immune features. The rationale has been added to the manuscript for better clarify and choice of NPC43.
The detection of the expression of BRRF1 in the siRNA experiments had been added to Figure S5. The detection of BRRF1-Myc was already shown in Figure 3B.
In terms of the suggestion to the title, it is a good suggestion to revise the title of the manuscript based on our data. Instead of referring to NPC in general, it would be more appropriate to specify the effect of BRRF1 on NPC43 cells. Therefore, we have revised the title to:
“Epstein‒Barr virus BRRF1 induces butyrophilin 2A1 expression in nasopharyngeal carcinoma NPC43 cells via the IL-22/JAK3-STAT3 pathway”
Cited references:
Liu Y, Lui KS, Ye Z, Fung TY, Chen L, Sit PY, Leung CY, Mak NK, Wong KL, Lung HL, Tanaka Y, Cheung AKL. EBV latent membrane protein 1 augments γδ T cell cytotoxicity against nasopharyngeal carcinoma by induction of butyrophilin molecules. Theranostics 2023 Jan 1;13(2):458-471. doi: 10.7150/thno.78395.
Lin, W.; Yip, Y.L.; Jia, L.; Deng, W.; Zheng, H.; Dai, W.; Ko, J.M.Y.; Lo, K.W.; Chung, G.T.Y.; Yip, K.Y.; et al. Establishment and characterization of new tumor xenografts and cancer cell lines from EBV-positive nasopharyngeal carcinoma. Nat Commun 2018, 9, 4663, doi:10.1038/s41467-018-06889-5.
Reviewer 2 Report (New Reviewer)
Comments and Suggestions for Authors
This work shows findings supporting that Epstein-Barr virus BRRF1 can induce butyrophilin 2A1 in nasopharyngeal carcinoma cells (NPC) via the IL-22/JAK3-STAT3 pathway.
The manuscript is well-written and organized. The data presented are convincing and novel.
It is not clear the functional consequences of the induction of BTN2A1 for the cell biology of NPC. There is no direct demonstration that the cells with a higher expression of BTN2A1 can better trigger Vdelta 2 T cell responses.
Also, it would be of interest to define whether NPC EBV infected could upregulate the expression of stressed molecules such as NKG2D ligands. Indeed, these molecules can concur with the triggering of Vdelta 2 T cells. This can be easily found by immunofluorescence assay and/or RT PCR analysis of NPC cells after the overexpression of BRRF1. The same analysis could involve molecules relevant for cell-to-cell binding.
Author Response
Reviewer 2
This work shows findings supporting that Epstein-Barr virus BRRF1 can induce butyrophilin 2A1 in nasopharyngeal carcinoma cells (NPC) via the IL-22/JAK3-STAT3 pathway.
The manuscript is well-written and organized. The data presented are convincing and novel.
Comment 1:
It is not clear the functional consequences of the induction of BTN2A1 for the cell biology of NPC. There is no direct demonstration that the cells with a higher expression of BTN2A1 can better trigger Vdelta 2 T cell responses.
Response 1:
Thank you for the comments. Although not shown in this manuscript, our recent study showed that the increased BTN2A1/BTN3A1 induced by the EBV EBNA1-targeting P4 could lead to an increased cytotoxicity exerted by Vd2 T cells in vitro (Liu et al, Theranostics 2023). In our NPC tumor model in NSG mice, adoptive transfer of Vd2 T cells result in the overall tumor regression. Upon tissue section analysis, the Vd2 T cells appear to infiltrate into the tumor cells that express the BTN2A1/BTN3A1 molecules, which may result in killing of these cells. This was also shown in a mesothelioma tumor mice model receiving Vd2 T cells (Lui et al., Frontiers in Immunology 2023).
Liu Y, Lui KS, Ye Z, Fung TY, Chen L, Sit PY, Leung CY, Mak NK, Wong KL, Lung HL, Tanaka Y, Cheung AKL. EBV latent membrane protein 1 augments γδ T cell cytotoxicity against nasopharyngeal carcinoma by induction of butyrophilin molecules. Theranostics. 2023 Jan 1;13(2):458-471. doi: 10.7150/thno.78395.
Lui KS, Ye Z, Chan HC, Tanaka Y, Cheung AKL*. Anti-PD1 does not improve pyroptosis induced by γδ T cells but promotes tumor regression in a pleural mesothelioma mouse model. Front Immunol. 2023 Nov 23;14:1282710. doi: 10.3389/fimmu.2023.1282710.
Comment 2:
Also, it would be of interest to define whether NPC EBV infected could upregulate the expression of stressed molecules such as NKG2D ligands. Indeed, these molecules can concur with the triggering of Vdelta 2 T cells. This can be easily found by immunofluorescence assay and/or RT PCR analysis of NPC cells after the overexpression of BRRF1. The same analysis could involve molecules relevant for cell-to-cell binding.
Response 2:
Thank you for the suggestions. Indeed, we found that the NKG2D ligands, namely ULBP3 and ULBP4, were upregulated in HK1-EBV and more so in the gastric tumor cell lines (NCC-24, SNU-719, AGS) overexpressed with BRRF1. Moreover, the adhesion molecules ICAM-1 and RAGE (Receptor for Advanced Glycation End-Products) were also upregulated among the cell lines tested except for SNU-719, and NCC-24/SNU-719, respectively. These data are now shown in Figure S6 and included in the Discussion.
Round 2
Reviewer 1 Report (Previous Reviewer 1)
Comments and Suggestions for Authors
The authors have reorganized their findings in order to generalize less, which is a good thing.
This manuscript is a resubmission of an earlier submission. The following is a list of the peer review reports and author responses from that submission.
Round 1
Reviewer 1 Report
Comments and Suggestions for Authors
The authors show that in NPC cells (NPC43 cells), overexpression of BRRF1 induces BTN2A1 expression. The proposed model is that overexpressed BRRF1 inhibits IL22RA2 expression, which may trigger activation of the BTN2A1 gene promoter via the JAK3-STAT3 pathway in NPC43 cells.
The manuscript suffers from several weaknesses and the results given are sometimes contradictory which makes the understanding of the text sometimes difficult.
1] The references given in the introduction concerning BRRF1 are not correct, they obviously refer to a subject other than the one covered.
2] The choice of NPC43 cells is unclear. Why are they more relevant than C666-1 cells? Were other EBV positive cells (epithelial cells or B cells) tested ?
3] Overexpression of BRRF1 appears to induce endogenous BTN2A1 and BTN3A1 expression in NPC43 cells (Figure 1A). However, if BRRF1 induces the BTN2A1 promoter cloned into a luciferase reporter, it appears to repress the BTN3A1 and NLRC5 promoters (Figure 1C). This difference in behaviour needs to be clarified and discussed.
4] Similarly, the results obtained for RhoB between the amount of mRNA and the amount of protein are different in the presence of BRRF1. This point needs to be discussed.
5] The proposed model is interesting, but needs to be supported by more arguments. The use of drugs provides a lead, but other controls could strengthen the hypothesis, for example, the use of siRNAs against factors in the Jack3/STAT3 pathway would seem necessary.
Reviewer 2 Report
Comments and Suggestions for Authors
The article titled “Epstein-Barr virus BRRF1 induces butyrophilin 2A1 in nasopharyngeal carcinoma cells via the IL-22/JAK3-STAT3 pathway” by Yue Liu, Zuodong Ye, Luo Chen and Allen Ka Loon Cheung examines the molecular mechanisms by which the virus Epstein-Barr virus BRRF1 gene -Barra (EBV) affects butyrophilin 2A1 (BTN2A1) expression in nasopharyngeal carcinoma (NPC) cells. This study is significant because it elucidates a novel mechanism by which EBV may modulate immune responses in NPC, potentially offering insight into immunotherapy.
Key findings and methodology:
BRRF1 induces BTN2A1 expression: Study shows that the EBV BRRF1 gene directly stimulates BTN2A1 expression in NPC43 cells. This finding is supported by RNA sequencing data and confirmed by inhibitor experiments.
Mechanisms involved in the JAK3-STAT3 pathway: BRRF1 was found to activate the JAK3-STAT3 signaling pathway in these cells, contributing to increased BTN2A1 expression. The involvement of this pathway was confirmed using specific inhibitors.
Downregulation of IL-22RA2: BRRF1 has also been shown to downregulate IL-22 binding protein (IL-22RA2), complementing IL-22 expression induced by the EBNA1-targeting P4 probe. This suggests a synergistic role of BRRF1 and IL-22 in modulating immune responses.
Experimental approach: This study used various molecular biology techniques, including RNA sequencing, RT-qPCR, Western blot, and luciferase reporter assays, to elucidate the mechanisms of action of BRRF1 in NPC cells.
The findings indicate that manipulating EBV genes, such as BRRF1, could potentially enhance immunotherapy against NPC. By understanding how EBV genes influence the expression of immune molecules, new strategies for the treatment of NPC can be developed. Study suggests a role for BRRF1 in possibly modifying tumors to improve the effectiveness of adoptive T-cell therapies.
Overall rating:
The study is well structured and uses robust methodologies to uncover new information on the interaction between EBV genetics and immune system modulation in nasopharyngeal cancer. It contributes significantly to the understanding of the pathogenesis of NPC and offers potential avenues for therapeutic intervention. Of particular note is the clear demonstration of a BRRF1-mediated mechanism and its implications for immunotherapy. Further research in this direction may lead to more effective treatments for patients with NPC.
Suggestions: It would be useful for the authors to discuss in more detail the limitations of their study and the potential impact of these limitations on the interpretation of the results. The authors can also expand the discussion section by specifically pointing out what next research steps can be taken to develop the field further.
Author Response
Authors: Thank you for the encouraging comments. Regarding the limitations, potential impact, interpretation of the results, and future directions were further elaborated on in the revised Discussion section.
Round 2
Reviewer 1 Report
Comments and Suggestions for Authors
Clearly, the authors refuse to take into account my comment on the relevance of the cell line used. To show that their results are a general observation, they have to show that they obtained the same results in different cell lines, at least of NPC origin. The argument that they don't have the time to do this is not valid in my opinion because this is essential. For the moment, they have an observation in a specific cell line. These results are too preliminary